# Selenium Yeast and Fish Oil Combination Diminishes Cancer Stem Cell Traits and Reverses Cisplatin Resistance in A549 Sphere Cells

**DOI:** 10.3390/nu14153232

**Published:** 2022-08-07

**Authors:** I-Chun Lai, Chien-Huang Liao, Ming-Hung Hu, Chia-Lun Chang, Gi-Ming Lai, Tzeon-Jye Chiou, Simon Hsia, Wei-Lun Tsai, Yu-Yin Lin, Shuang-En Chuang, Jacqueline Whang-Peng, Hsuan-Yu Chen, Chih-Jung Yao

**Affiliations:** 1The Ph.D. Program for Translational Medicine, College of Medical Science and Technology, Taipei Medical University and Academia Sinica, Taipei 11031, Taiwan; 2Division of Radiation Oncology, Department of Oncology, Taipei Veterans General Hospital, Taipei 11217, Taiwan; 3School of Medicine, National Yang Ming Chiao Tung University, Taipei 11230, Taiwan; 4Cancer Center, Wan Fang Hospital, Taipei Medical University, Taipei 11696, Taiwan; 5Division of Hematology and Medical Oncology, Department of Internal Medicine, Wan Fang Hospital, Taipei Medical University, Taipei 11696, Taiwan; 6Department of Internal Medicine, School of Medicine, College of Medicine, Taipei Medical University, Taipei 11031, Taiwan; 7Taiwan Nutraceutical Association, Taipei 10596, Taiwan; 8National Institute of Cancer Research, National Health Research Institutes, Miaoli 35053, Taiwan; 9Ph.D. Program for Translational Medicine, National Taiwan University, Taipei 10051, Taiwan; 10Genome and Systems Biology Degree Program, National Taiwan University, Taipei 10051, Taiwan; 11Ph.D. Program in Microbial Genomics, National Chung Hsing University, Taichung 40227, Taiwan; 12Department of Medical Research, Kaohsiung Medical University Hospital, Kaohsiung 80761, Taiwan; 13Institute of Statistical Science, Academia Sinica, Taipei 11529, Taiwan; 14Department of Medical Education and Research, Wan Fang Hospital, Taipei Medical University, Taipei 11696, Taiwan

**Keywords:** selenium, lung cancer, cisplatin, fish oil, cancer stem cell, AMPK, side population

## Abstract

Cisplatin is a prevalent chemotherapeutic agent used for non-small cell lung cancer (NSCLC) that is difficult to treat by targeted therapy, but the emergence of resistance severely limits its efficacy. Thus, an effective strategy to combat cisplatin resistance is required. This study demonstrated that, at clinically achievable concentrations, the combination of selenium yeast (Se-Y) and fish oil (FO) could synergistically induce the apoptosis of cancer stem cell (CSC)-like A549 NSCLC sphere cells, accompanied by a reversal of their resistance to cisplatin. Compared to parental A549 cells, sphere cells have higher cisplatin resistance and possess elevated CSC markers (CD133 and ABCG2), epithelial–mesenchymal transition markers (anexelekto (AXL), vimentin, and N-cadherin), and cytoprotective endoplasmic reticulum (ER) stress marker (glucose-regulated protein 78) and increased oncogenic drivers, such as yes-associated protein, transcriptional coactivator with PDZ-binding motif, β-catenin, and cyclooxygenase-2. In contrast, the proapoptotic ER stress marker CCAAT/enhancer-binding protein homologous protein and AMP-activated protein kinase (AMPK) activity were reduced in sphere cells. The Se-Y and FO combination synergistically counteracted the above molecular features of A549 sphere cells and diminished their elevated CSC-like side population. AMPK inhibition by compound C restored the side population proportion diminished by this nutrient combination. The results suggest that the Se-Y and FO combination can potentially improve the outcome of cisplatin-treated NSCLC with phenotypes such as A549 cells.

## 1. Introduction

Lung cancer is the leading cause of cancer-related mortality worldwide, accounting for approximately eighteen percent (1.8 million) of all cancer deaths [1]. Approximately 85% of lung cancers are non-small cell lung cancers (NSCLCs), including adenocarcinoma, squamous cell carcinoma, and large cell carcinoma [2]. The most dominant subtype is lung adenocarcinoma [3]. Despite recent advances in the effectiveness of tyrosine kinase inhibitors and vast chemotherapeutics, drug resistance often occurs after the initial effective treatment, resulting in relapse and fatal outcomes. Evidence indicates the critical contribution of cancer stem cells (CSCs) to drug resistance and eventual treatment failure. Therefore, effective therapeutics to eliminate CSC is urgently required. However, most conventional anticancer drugs fail to effectively eradicate CSCs [4], and some of them have been reported to foster CSC traits [2,5,6,7]. On the contrary, diverse natural products, including diet-derived nutrients, are known to suppress CSC traits and related signaling molecules, representing a cornucopia of CSC inhibitors [4,8,9]. Nonetheless, most natural products have modest potency and may not be used as a single agent for satisfactory anticancer effects [9]. Therefore, the combination of natural products for more potent anticancer efficacy has been suggested [10,11,12,13].

Selenium yeast (Se-Y) and fish oil (FO) are well known as nutrients, and their individual anticancer activities have been widely examined. However, their clinical efficacies are still superficially plausible, and certain clinical trial outcomes are not substantially satisfactory or are even conflicting [14,15]. Remarkably, a previous study reported the synergistic combined effects of Se-Y and FO on apoptosis induction in A549 lung adenocarcinoma cells via AMP-activated protein kinase (AMPK) activation and the opposite regulation of proapoptotic CCAAT/enhancer-binding protein homologous protein (CHOP) and cytoprotective glucose-regulated protein 78 (GRP78) endoplasmic reticulum (ER) stress-response elements, accompanied by a decrease in β-catenin and cyclooxygenase-2 (COX-2) [16]. These targeted molecules (CHOP [17], GRP78 [17,18], β-catenin [19], and COX-2 [20]) are closely associated with gefitinib resistance in lung cancer. Hence, subsequent study employed the Se-Y and FO combination to reverse acquired gefitinib resistance of lung adenocarcinoma HCC827 cells (E746-A750 exon 19 deletion) with elevated GRP78, β-catenin, COX-2, and CSC markers (ABCG2 and CD133), and epithelial–mesenchymal transition (EMT) markers (vimentin and anexelekto (AXL)) [21]. Besides the association with acquired gefitinib resistance [21], the aforementioned CSC and EMT markers have been reported to be involved in the resistance to the widely prescribed chemotherapy agent cisplatin [5,6,22,23,24,25,26]. The suppressing effects of the Se-Y and FO combination on elevated CSC and EMT markers in gefitinib-resistant HCC827 cells [21] imply the capability of reversing cisplatin resistance in lung CSCs. Additionally, AMPK suppression elevates the ATP-dependent efflux pump ABCG2 in A549 cells, increasing the side population of chemoresistant cells [7]. Activation of AMPK by the Se-Y and FO combination in A549 cells [16] is proposed to eliminate the above-described CSC-like side population.

To investigate the valuable potentiality of this nutrient combination, CSC-like A549 sphere cells were enriched and collected by culturing A549 lung adenocarcinoma cells [epidermal growth factor receptor (EGFR) wild-type, KRAS-G12S] as spheroid bodies in a defined small-molecule-based serum-free medium. Consistent with the reported characteristics of sphere cells [27,28,29], these A549 sphere cells displayed higher CSC and EMT markers and elevated cisplatin resistance. This study explored the combined effects of Se-Y and FO on A549 sphere cells, including CSC trait diminishment, cisplatin resistance reversion, apoptosis induction, ER stress modification, and AMPK activation. Moreover, AMPK inhibits yes-associated protein (YAP) and its paralog, transcriptional coactivator with PDZ-binding motif (TAZ), which are oncogenic drivers conferring CSC traits [30]. The combined effects of Se-Y and FO on YAP and TAZ in CSC-like A549 sphere cells were also studied.

## 2. Materials and Methods

### 2.1. Cell Culture

The A549 human NSCLC cell line was obtained from Bioresource Collection and Research Center (Hsinchu, Taiwan) and cultured in RPMI-1640 medium (Gibco, CA, USA) supplemented with 10% fetal bovine serum (35-010-CV, Corning Incorporated, Corning, NY, USA), 1× penicillin–streptomycin–glutamine (Corning Incorporated, Corning, NY, USA), and 1× nonessential amino acids (Corning Incorporated, Corning, NY, USA). To obtain sphere cultures, A549 cells were seeded at a density of 2.24 × 10^7^ cells/10 cm culture dish (150466, Thermo Fisher Scientific, Nunc, Waltham, MA, USA) containing 8 mL PluriSTEM™ human ES/iPS Cell medium (SCM130, Millipore, Burlington, MA, USA) supplemented with 1× penicillin–streptomycin–glutamine. After 24 h, the attached cells were discarded and the floating cells with round and smooth contour were collected, transferred to an uncoated 10 cm dish (70165-102, Corning, Oneonta, NY, USA) at density of 2 × 10^6^ cells/dish and cultured with 8 mL PluriSTEM™ human ES/iPS cell medium for another 24 h. In the following 5 days, the growing cells were split 1:1 to expand the number. To keep the conditioned medium, only half of the medium was refreshed upon splitting. The clumping of cells was resolved by trituration without using trypsin/ethylenediaminetetraacetic acid (EDTA). After being cultured for 7 days, the sphere cells were analyzed for their CSC traits and then used for subsequent experiments. Except as specifically depicted in the legends of Figures in Results section, the experiments of A549 sphere cells were performed in uncoated 10 cm dishes with 8 mL PluriSTEM™ human ES/iPS cell medium. The RPMI-1640 medium was supplemented with 10% fetal bovine serum, 1× penicillin–streptomycin–glutamine and 1× nonessential amino acids, and the PluriSTEM™ human ES/iPS cell medium was supplemented with 1× penicillin–streptomycin–glutamine. All cells were maintained in a water-jacketed 5% carbon dioxide (CO_2_) incubator at 37 °C.

### 2.2. Reagents and Chemicals

Se-Y was employed as the form of selenium for the treatment of NSCLC cells. The stated concentration of Se-Y indicates the content of selenium from selenium yeast. The stock solutions of Se-Y and fish oil (FO, each gram contained 220 mg of docosahexaenoic acid (DHA) and 330 mg of eicosapentaenoic acid (EPA)) were obtained from Dr. Chih-Hung Guo (Institute of Biomedical Nutrition, Hung-Kuang University, Taichung, Taiwan). In brief, Se-Y is a mixture of small-molecule peptide-bound selenium extracted from yeast grown in selenium-enriched medium. FO is produced from anchovy by supercritical CO_2_ extraction. FO is a mixture of natural triglyceride (TG) form of omega-3 fatty acids. They were then aliquoted, stored at −80 °C (Se-Y) and −20 °C (FO), and diluted in sterile culture medium immediately prior to use. The concentrations of FO depicted in text and figures represent the content of omega-3 fatty acids (DHA and EPA). When combined with cisplatin, Se-Y and FO were added to A549 sphere cells and then the cisplatin was added. Cisplatin was obtained from clinical preparations of Abiplatin solution (Pharmachemie BV, Haarlem, the Netherlands). DyeCycle Violet (DCV) dye was purchased from Invitrogen (Invitrogen, Carlsbad, CA, USA). Sulforhodamine B (SRB), compound C (also called dorsomorphin, an AMPK inhibitor), propidium iodide, and trichloroacetic acid were bought from Sigma-Aldrich (St. Louis, MO, USA). FITC Annexin V Apoptosis Detection Kit with 7-amino-actinomycin D (7-AAD) was bought from BioLegend Inc. (San Diego, CA, USA).

### 2.3. Images of the Cells

A digital microscope camera PAXcam2+ (Midwest Information Systems, Inc., Villa Park, IL, USA) adapted with an inverted microscope CKX31 (Olympus Co., Tokyo, Japan) was used to take the images of cell cultures in Figure 1A (at 10× objective lens magnification).

### 2.4. Colony Formation Assay

A549 cells and cultivated A549 sphere cells were plated at density of 150 cells/well in 6-well plate (353046, Falcon, Corning, NY, USA) with RPMI-1640 medium. The formation of colonies was photographed after 7 days of culture. Crystal violet (2%) (C0775, Sigma-Aldrich, Merck KGaA, Darmstadt, Germany) dissolved in 20% methanol solution was used to fix and stain the colonies.

### 2.5. Side Population Detection by DyeCycle Violet (DCV) Exclusion

Twenty four hours after being plated in 10 cm dish (3 × 10^5^ parental A549 cells/dish, 1 × 10^6^ A549 sphere cells/dish), agents were added to cells as described in the figure legends. After treatment, cells were harvested from the culture dish, washed twice with phosphate-buffered saline (PBS), and resuspended at a concentration of 1 × 10^6^ cells/mL in culture medium at 37 °C (parental A549 in RPMI-1640 and A549 sphere in PluriSTEM™ human ES/iPS cell medium). While protected from light, these cells (1 mL, 1 × 10^6^) were then incubated for 90 min at 37 °C with either 1 μL (final stain concentration is 5 μM) of DCV alone or in combination with verapamil or reserpine as depicted in the figures. Subsequently, cells were centrifuged immediately for 5 min at 240× *g* and 4 °C, and then resuspended in ice-cold PBS. Propidium iodide (final concentration of 50 μg/mL) was then added to discriminate dead cells. The cells were then kept at 4 °C until subjected to flow cytometric analysis at an excitation of 405 nm and emissions of 450 and 660 nm using a CytoFLEX flow cytometer (Beckman Coulter, Inc., Indianapolis, IN, USA). The gating of side population was confirmed by reserpine and verapamil (ATP-binding cassette (ABC) transporter inhibitors), as indicated.

### 2.6. Apoptotic and Necrotic Cell Death Detected by Annexin V/7-Amino-Actinomycin D Double Staining

After being plated in a 10 cm dish (3 × 10^5^ parental A549 cells/dish, 1 × 10^6^ A549 sphere cells/dish) for 24 hours, cells were treated with agents as depicted in the figures for 3 days. Apoptosis and necrosis induction in treated cells were examined by double staining of annexin V and 7-amino-actinomycin D (7-AAD). Double staining with annexin V and 7-AAD is commonly used to discriminate early apoptosis from late apoptosis and necrosis. In early apoptotic stages, cells can exclude the vital dyes, such as 7-AAD (DNA intercalator) because they still maintain the plasma membrane integrity, whereas the phosphatidylserines (PS) on the outer leaflet of the plasma membrane will be stained by annexin V labeled with fluorescein. In contrast, late apoptotic and necrotic cells can be stained by 7-AAD as they lose cell membrane integrity [31]. At harvest, cold Cell Staining Buffer (BioLegend, Inc., San Diego, CA, USA) was used to wash the cells twice, and they were resuspended in Annexin V Binding Buffer (BioLegend, Inc., San Diego, CA, USA) at a density of 1.6 × 10^6^ cells/mL. First, 100 µL of the cell suspension was transferred to a 5 mL test tube and then 5 µL of FITC Annexin V (BioLegend, Inc., San Diego, CA, USA) and 5 µL of 7-AAD Viability Staining Solution (BioLegend, Inc., San Diego, CA, USA) were added. After being gently vortexed and incubated for 15 min at room temperature (25 °C) while protected from light, 400 µL of Annexin V Binding Buffer (BioLegend, Inc., San Diego, CA, USA) was added to the tube and then the cells were subjected to flow cytometric analysis by CytoFLEX (Beckman Coulter, Inc., Indianapolis, IN, USA). At least ten thousand events were collected and analyzed.

### 2.7. Measurement of Apoptotic Sub-G1 Fraction

Twenty-four hours after being cultured in 10-cm dish (3 × 10^5^ parental A549 cells/dish, 1 × 10^6^ A549 sphere cells/dish), cells were treated with agents as depicted in the figure legends. After treatment, the cells were centrifuged. The cell pellets were reacted with 500 µL (2 µg/mL) of DAPI (4’,6-diamidino-2-phenylindole) solution (NPE 731085, Beckman Coulter, Fullerton, USA) in a 1.5 mL Eppendorf tube on ice and then the cell-cycle distribution and apoptotic sub-G1 percentage were measured by a CytoFLEX flow cytometer (Beckman Coulter, Inc., Indianapolis, IN, USA). At least ten thousand events were collected and analyzed.

### 2.8. Western Blot

After being cultured in a 10 cm dish (3 × 10^5^ parental A549 cells/dish, 1 × 10^6^ A549 sphere cells/dish) for one day, cells were then treated with agents as depicted in the figures. After treatment, the whole-cell lysates were extracted with 1× radioimmunoprecipitation lysis buffer (Merck Millipore, Billerica, MA, USA) containing 1× serine/threonine phosphatase inhibitor cocktail (FC0030-0001, BIONOVAS, Toronto, ON, Canada), 1× tyrosine phosphatase inhibitor cocktail (FC0020-0001, BIONOVAS, Toronto, ON, Canada), and 1× protease inhibitor cocktail, full range (FC0070-0001, BIONOVAS, Toronto, ON, Canada). The protein extracts were separated by sodium dodecyl sulfate–polyacrylamide gel electrophoresis (SDS–PAGE) and then transferred to polyvinylidene difluoride (PVDF) membrane (GE Healthcare, Pittsburgh, PA, USA) by electroblotting. The membranes were blocked by bovine serum albumin (5%) in Tris-buffered saline (TBST) buffer (Tris-buffered saline with Tween 20, 25 mM Tris–hydrochloric acid (HCl), 125 mM sodium chloride (NaCl), 0.1% Tween 20) at room temperature for 1 h and incubated with primary antibody at 4 °C overnight and then with secondary antibody (horseradish peroxidase-conjugated) at room temperature for 1 h. After each incubation period, the membrane was intensively washed with TBST buffer. Finally, the target proteins of the primary antibodies were visualized using enhanced chemiluminescence (ECL) Reagent Plus (Perkin Elmer, Inc., Waltham, MA, USA) on the Syngene G:Box chemi XL gel documentation system (Syngene, Cambridge, UK) according to the manufacturer’s instructions. The intensities of Western blot bands were quantified by using ImageJ software (ImageJ Version 1.52, National Institutes of Health, Bethesda, MD, USA) downloaded from https://imagej.nih.gov/ij/download.html (accessed on 25 September 2019).

### 2.9. Antibodies

Primary antibodies against non-phospho (active) β-catenin (Ser33/37/Thr41, #8814), ABCG2 (#4477), phospho-AMPKα (Thr172, #2535), AXL (#8661), TAZ (#8418), active caspase-4 (#4450), full-length caspase-8 (#9746), cleaved caspase-3 (Asp175, #9664), and cleaved caspase-9 (Asp315, #9505) were purchased from Cell Signaling Technology, Inc. (Danvers, MA, USA). Primary antibodies for CHOP (ab11419), GRP78 (ab108613), COX-2 (Ab62331), CD133 (ab19898), vimentin (ab16700), N-cadherin (ab76011), YAP (ab52771), Active YAP (ab205270), Bcl-2 (ab32124), and glyceraldehyde-3-phosphate dehydrogenase (GAPDH) (Ab8245) were bought from Abcam, Inc. (Cambridge, MA, USA).

### 2.10. Cell Viability Measurement

The A549 sphere cells were plated (2000 cells/well) in 96-well culture plates (167008, Thermo Fisher Scientific, Nunc, Waltham, MA, USA) with RPMI-1640 medium for 24 h and then treated with agents indicated in the figures for 3 days. SRB binding assay was used to measure the cell viability. In brief, the cells were fixed by trichloroacetic acid (10%) and incubated for 1 h at 4 °C. Then, the plates were washed with tap water twice and dried in air. The dried plates were stained with 80 µL of 0.4% (*w/v*) SRB prepared in 1% (*v/v*) acetic acid at room temperature for 30 min. The unbound SRB in plates was removed by two quick rinses with 1% acetic acid, and the plates were then dried in air until no moisture was apparent. Finally, 20 mmol/L Tris base (200 µL/well) was added to dissolve the bound dye on a shaker for 5 min. Optical density at 570 nm was read by a microplate reader ELx800 (BioTek Instruments, Inc., Winooski, VT, USA). The optical density is directly proportional to the viable number of cells over a wide range.

### 2.11. Analysis of Synergistic Combination Effect

The synergistic effect between two treatments on the cell viability of sphere cells was evaluated by the mutually non-exclusive combination index (CI) calculated from the median effect principle of Chou and Talalay [32], using the CalcuSyn software (version 1.1.1; Biosoft, Cambridge, UK). A value of CI = 1 means an additive effect, whereas values of CI < 1 or CI > 1 mean synergistic or antagonistic effect, respectively.

## 3. Results

### 3.1. CSC-like A549 Sphere Cells Possessed Elevated GRP78 and Reduced CHOP and AMPK Activity

After culture in a defined small-molecule-based serum-free medium in a nonadhesive culture system, floating spheres with round and smooth contours were successfully obtained from A549 (EGFR wild-type, KRAS-G12S) human lung adenocarcinoma cells (Figure 1A). To characterize their stemness, the expression of lung CSC markers, such as CD133 [2] and ABCG2 [33], was examined. Consistent with the previously reported culture of sphere-growing lung cancer cells [34], these A549 sphere cells were highly enriched for CD133 than parental A549 cells (Figure 1B). Consistent with enriched CD133+ cells from human lung cancer cell lines [2], coexpression of elevated ABCG2 was observed in these spheres (Figure 1B). Besides ABCG2, the level of another chemoresistance driver, B-cell lymphoma-2 (Bcl-2) [35], in these sphere cells was also higher than in parental cells (Figure 1B). In accordance with the functional and mechanistic links between EMT and cancer stemness [36], the levels of EMT markers such as vimentin, N-cadherin, and AXL [37] were higher in sphere cells (Figure 1B,C). Moreover, the CSC traits of these sphere cells were further characterized by the elevated YAP and TAZ oncogenic driver levels (Figure 1C). Next, this study examined the reported targets of selenium (AMPK, β-catenin, and COX-2) [38,39] and the ER stress-response elements (GRP78 and CHOP) that could be synergistically regulated by the Se-Y and FO combination as reported in previous studies [16,21]. As shown in Figure 1D, A549 sphere cells have significantly lower AMPK activity and CHOP levels than parental cells and higher levels of β-catenin, COX-2, and GRP78. These molecular features are paradoxical to the combined effects of Se-Y and FO in previous studies [16,21], implying the potential utility of this nutrient combination to counteract these molecular features, accompanied by diminishing the aforementioned CSC traits and cisplatin resistance.

A549 sphere cells were analyzed for clonogenicity, side population proportion, and cisplatin resistance. Consistent with the reported proliferation capability of sphere cells from A549 cells [28], A549 sphere cells in this study formed a larger and higher number of colonies than parental A549 cells after culture for 7 days (Figure 2A). Similar to the elevated ABCG2, the side population percentage increased from 3.57% in parental cells to 19.4% in sphere cells (Figure 2B).

When cells were treated with cisplatin (20 μg/mL) for 72 h, the apoptotic sub-G1 fraction in parental A549 cells was markedly increased, from 1.64% to 24.77% (Figure 2C, top). In contrast, the sub-G1 fraction in floating A549 sphere cells only slightly increased, from 8.58% to 10.1%, at the same concentration of cisplatin (Figure 2C, middle). After treating A549 sphere cells with cisplatin (20 μg/mL) in the same serum containing culture condition as parental cells, the sub-G1 fraction still slightly increased from 6.13% to 8.19% (Figure 2C, bottom). The cell cycle distribution of these cisplatin (20 μg/mL)-treated cells is described in Table 1. This phenomenon was further confirmed by the annexin V/7-AAD double staining assay that revealed compatible percentages of early apoptosis (lower right quadrant), late apoptosis (upper right quadrant), and necrosis (upper left quadrant) of cells (Figure 2D) treated as that in Figure 2C. Compared to parental cells, A549 sphere cells were relatively resistant to cisplatin-induced cell death.

### 3.2. Se-Y and FO Synergistically Induced Apoptosis of A549 Sphere Cells and Diminished CSC Traits

This study assessed the combined effects of Se-Y and FO on cisplatin-resistant A549 sphere cells. Consistent with a previous study [16], the combination of Se-Y (200 ng/mL) and FO (200 μM) induced the apoptotic sub-G1 fraction from 1.9% in the control group to 42.2%, whereas individual treatment only slightly increased the apoptotic fraction, to 14.4% and 3.4%, respectively (Figure 3A). The combined effects of Se-Y and FO on the cell-cycle distribution of A549 sphere cells are listed in Table 2. This result was further confirmed by the annexin V/7-AAD double staining assay, which showed compatible percentages of early apoptosis (lower right quadrant), late apoptosis (upper right quadrant) and necrosis (upper left quadrant) of A549 sphere cells (Figure 3B) treated as that in Figure 3A. Accordingly, the cell viability assay showed that combination of Se-Y (200 ng/mL) and FO (200 μM) suppressed the viability of A549 sphere cells to 27.3% of control, whereas individual treatment only slightly reduced the cell viability to 70.3% and 77.9% of control, respectively (Figure 3C). Evaluating the synergism of this combination by combination index (CI), most of the CI values were all below 1 (synergistic effect), except a value of 1.078 (Table 3). Further, the combined effects on ER stress-response elements (CHOP and GRP78) and the reported targets of selenium (β-catenin and COX-2) [38,39] were examined. Although the individual effects of Se-Y (200 ng/mL) and FO (200 μM) were relatively mild, the combination of these nutrients not only reduced the elevated prosurvival GRP78 level, but also drastically increased the proapoptotic CHOP of A549 sphere cells (Figure 3D), along with marked activation of caspase-3, the terminal executioner of apoptosis (Figure 3E). A similar pattern of the combined effects on reducing β-catenin and COX-2 was also observed in A549 sphere cells treated with the Se-Y and FO combination (Figure 3D). In agreement with the proapoptotic modulation of GRP78 and CHOP (Figure 3D), ER stress-related caspase-4 [40] in A549 sphere cells was synergistically activated by the combination of Se-Y and FO (Figure 3E). Similar effects were also observed on caspase-9 induction, antiapoptotic Bcl-2 suppression, and full-length caspase-8 cleavage (Figure 3E).

The protein levels of lung CSC markers, such as CD133 [2] and ABCG2 [33], were examined to determine whether this synergistic apoptosis induction was accompanied by the suppression of CSC traits. As expected, elevated ABCG2 and CD133 in A549 sphere cells were suppressed by the Se-Y and FO combination (Figure 4A). In agreement with its reported effect on AMPK activation [16], this combination increased AMPK activity upon suppressing ABCG2 in A549 sphere cells (Figure 4A). Additionally, elevated EMT markers (vimentin, AXL, and N-cadherin) and oncogenic drivers conferring CSC traits (active YAP, YAP, and its paralog TAZ) were all suppressed by the Se-Y and FO combination in a similar manner (Figure 4B). The combination of Se-Y and FO not only induced apoptosis, but also diminished CSC traits and EMT markers of cisplatin-resistant A549 sphere cells.

### 3.3. Se-Y and FO Combination Suppressed the Side Population of A549 Sphere Cells Via AMPK Activation

AMPK activation suppressed ABCG2 in A549 cells and thus decreased the side population percentage [7]. Subsequently, the side population percentage in A549 sphere cells was analyzed after AMPK modulation by the Se-Y and FO combination or AMPK inhibitor compound C (dorsomorphin). In Figure 5A, AMPK activation by the Se-Y and FO combination was eliminated when sphere cells were pretreated with compound C for 1 h and washed out. In line with the modulation of AMPK activity, the side population percentage was reduced from 16.1% in control sphere cells to 5.9% by the combination of Se-Y and FO (Figure 5B); in contrast, in sphere cells pretreated with compound C for 1 h and washed, the side population was only reduced to 11.5% by the combination of Se-Y and FO (Figure 5B). There was a negative reciprocal interplay between AMPK and the side population proportion in A549 sphere cells.

### 3.4. Se-Y and FO Combination Reversed Cisplatin Resistance in A549 Sphere Cells

In addition to inducing the side population phenotype [7], the efflux molecule ABCG2 also contributes to A549 cell resistance to cisplatin [22,23,41]. Given the substantial decrease of ABCG2 in this nutrient combination-treated A549 sphere cells, their cisplatin sensitivity was analyzed. As expected, cisplatin (20 μg/mL) did not substantially increase the apoptotic sub-G1 fraction (9.37% to 8.34%) in A549 sphere cells after 72 h of treatment (Figure 6A). However, cisplatin (20 μg/mL) further increased the sub-G1 fraction in Se-Y (200 ng/mL) and FO (200 μM) combination-treated sphere cells from 45.87% to 63.02% (Figure 6A). The Se-Y and FO combination at half the concentrations in Figure 6A elevated the sub-G1 fraction of A549 sphere cells from 8.46% to 37.77%. This apoptotic fraction could be further enhanced to 71.58% by cisplatin (20 μg/mL; Figure 6B). These data indicate that combining Se-Y and FO restored A549 sphere cell sensitivity to cisplatin. Table 4 and Table 5 list the cell-cycle distribution of A549 sphere cells described in Figure 6A,B, respectively. The results in Figure 6B were further confirmed by the annexin V/7-AAD double staining assay showing compatible percentages of early apoptosis (lower right quadrant), late apoptosis (upper right quadrant), and necrosis (upper left quadrant) of A549 sphere cells (Figure 6C) treated as in Figure 6B. In the presence of the Se-Y and FO combination, A549 sphere cell resistance to cisplatin was reversed. A similar result was also obtained by the cell viability test determined by sulforhodamine B staining. As shown in Figure 6D, the effect of cisplatin on the cell viability of A549 sphere cells was enhanced by the combination of Se-Y and FO. When evaluating the synergism by CI, the CI values of the combination of cisplatin with Se-Y and FO were all below 1 (Table 6), which indicates synergistic effect.

## 4. Discussion

Cisplatin resistance remains a serious unresolved clinical and scientific challenge in lung cancer treatment. Apart from the tyrosine kinase inhibitor (TKI)-responsive population, NSCLC patients without mutations for targeted therapies remain heavily reliant on conventional chemotherapy, in which cisplatin is primarily used. Although a successful initial response is achieved in some patients, drug resistance eventually develops in all lung cancers [28]. Cisplatin resistance was detected in 1409 (63%) of 2227 NSCLC clinical cell cultures, which may account for the marginal survival advantage after empiric adjuvant chemotherapy for resected NSCLC [42]. Only a minor portion of NSCLC patients have an improved survival advantage after adjuvant chemotherapy, and most endure toxicity without acquiring benefits [42]. Thus far, no clinically available drug can overcome cisplatin resistance or kill cisplatin-resistant cells [43]. Novel therapeutics or alternative strategies are pressingly needed. This study demonstrated the synergistic effects of combining Se-Y and FO on apoptosis induction in cisplatin-resistant A549 sphere cells, accompanied by diminishing CD133 and ABCG2, two important therapeutic targets for relieving the cisplatin resistance of NSCLC. CD133 protein expression tended to correlate with a shorter median progression-free survival and early recurrence in stage IIIB/IV NSCLC patients treated with platinum-containing regimens [6]. ABCG2 overexpression confers the side population phenotype, which is employed for quantifying the chemoresistant subpopulation in cancer cells [7]. ABCG2 inhibition resensitized NSCLC to cisplatin [22,41]. Accordingly, CD133+ABCG2+ subpopulation cells could be spared from cisplatin treatment in mice xenografts established from human primary lung cancers [6]. In agreement with mentioned above, substantial inhibition of ABCG2 and CD133 in A549 sphere cells by the Se-Y and FO combination resulted in a smaller side population and cisplatin resistance.

There was a negative reciprocal relationship between AMPK activity and the side population percentage of parental and sphere-forming populations of A549 cells, and the Se-Y and FO combination could activate AMPK to reduce the side population of A549 sphere cells. Liu et al. demonstrated that increasing glucose concentration could suppress AMPK to expand the CSC-like side population in cell lines, including A549 [7]. They demonstrated that cisplatin mainly killed non-side population cells and thus increased the side population and tumor-initiating capacity (in mice) of A549 cells [7]. In contrast, the glycolysis inhibitor 3-bromo-2-oxopropionate-1-propyl ester (3-BrOP) could activate AMPK and reduce the side population in A549 cells, resulting in impaired tumor-initiating capacity in mice [7]. The elimination of the side population appears to be a potentially effective strategy for improving NSCLC treatment. In line with this, combining Se-Y and FO reduced the side population in A549 sphere cells and reversed their cisplatin resistance. In the study conducted by Liu et al., the high-glucose–elevated CSC-like side population [7] might echo studies illustrating that hyperglycemia may contribute to a more malignant phenotype of cancer cells and confer resistance to chemotherapy [44,45]. In addition to diabetes mellitus, hyperglycemia can occur in obesity, pancreatitis, chronic stress, and cancer [44]. In cancer patients harboring phenotypes as A549 cells and the above co-morbidities, the combination of Se-Y and FO might potentially alleviate hyperglycemia-mediated cancer fostering effects by diminishing the elevated CSC-like side population.

AMPK is an emerging anticancer target [46]. Accordingly, this study and the one by Liu et al. [7] showed a negative reciprocal interaction between AMPK activity and the side population. Thus, metformin, a well-known antidiabetic drug and an AMPK activator, has recently received a surge of interest for its potential as an anticancer agent [46]. Metformin was shown to relieve cisplatin resistance [47], eliminate CSCs [48,49,50], and provide benefits in cancer prevention and treatment [45,46,49,51]. However, unlike the clinically achievable concentrations of Se-Y and FO [52,53] used in this study, most of the effective anticancer concentrations of metformin employed in preclinical studies were supraphysiological (at the millimolar range) [50,54]. Although metformin penetrated certain tumors and concentrations reaching 100 μM were detected in rodents [55], the maximum plasma concentration of metformin in healthy human subjects was only 8–16 μM [56]. Considering the combined effects of Se-Y and FO at clinically achievable concentrations, it seemed logical to postulate that combining Se-Y and FO might exert benefits for cancer prevention and treatment similar to metformin, including but not limited to the condition of hyperglycemia.

AMPK inhibits the oncogenic driver YAP and its paralog TAZ, which regulate CSC biology, including drug resistance, EMT [30], ABCG2, and the side population [57]. In agreement, compared to parental A549 cells, AMPK activity was reduced in sphere cells. YAP activity and the related CSC and EMT markers were elevated. Consistent with AMPK activation by the combination of Se-Y and FO, the profound inhibition of YAP activity was displayed in these treated A549 sphere cells, and the features of YAP induction were substantially reversed.

Besides YAP, the aberrant elevation of the cytoprotective ER chaperone GRP78 also promoted the CSC traits, EMT, drug resistance, metastasis, and tumorigenesis of lung cancer cells [58]. Accordingly, elevated GRP78 was observed in CSC-like A549 sphere cells. Their lowered level of proapoptotic ER stress effector CHOP further confirmed their cisplatin resistance. Unlike the simultaneous induction of CHOP and GRP78 by selenium (methylseleninic acid) alone in prostate cancer cells [59], a previous study reported the opposite regulation of CHOP and GRP78 by the combination of Se-Y and FO to induce apoptosis in A549 NSCLC cells via AMPK activation [16]. Consistent with observations in gefitinib-resistant HCC827 NSCLC cells [21], this study further demonstrated CHOP induction along with GRP78 suppression by the Se-Y and FO combination in cisplatin-resistant A549 sphere cells. The ER stress-related caspase-4 [40] in these treated sphere cells was synergistically activated upon massive induction of apoptosis. GRP78 is considered a potential marker for lung cancer diagnosis and prognosis [58]. Besides the above-mentioned AMPK [46] and ABCG2 [6], GRP78 is also an emerging anticancer target for new therapeutic interventions [58]. Targeting cell-surface GRP78 enhanced pancreatic cancer radiosensitivity by inhibiting YAP/TAZ protein signaling [60]. The inhibition of GRP78 and YAP/TAZ by the Se-Y and FO combination suggested its potential radiosensitizing effect in these treated A549 sphere cells. In accordance with this, the nutrition supplement containing Se-Y, EPA, and DHA enhanced the anticancer effects of radiotherapy on lung cancer-bearing mice [61]. The GRP78-suppressing effects of the Se-Y and FO combination in previous studies [16,21] and this study also suggested the potential clinical impact of this nutrient combination in lung cancer therapy.

Recent studies supported the notion that AMPK activation enhances the anticancer effects of cisplatin [62,63]. Nevertheless, some studies reported conflicting results on the tumor-promoting effects of AMPK activation [64,65]. Now, it is proposed that AMPK can behave as a CSC “friend” or “foe” in a tissue type-specific and context-dependent manner [65]. The widespread use of AMPK activators should be trialed cautiously across disease settings of cancers [64]. Cancer cells elicit complicated responses that are not fully understood to survive from treatments. Treatment with cisplatin combined with Se-Y and FO might also lead to a resistant subpopulation characterized by different responses to AMPK activation. Nevertheless, based on this study and the reported tumor-suppressing effects of AMPK activation in A549 cells [7,63], the combination of Se-Y and FO may potentially improve the outcomes of cisplatin-treated NSCLC with phenotypes such as A549 cells and warrant further tests in preclinical mouse models, followed by clinical evaluation.

## 5. Conclusions

Cisplatin is widely prescribed for NSCLC that is difficult to treat by targeted therapy. Studies worldwide have explored therapeutics to overcome NSCLC cisplatin resistance. This study demonstrated that combining Se-Y and FO at clinically achievable concentrations could synergistically induce apoptosis in A549 NSCLC sphere cells, accompanied by a reversal of their cisplatin resistance. As shown in Figure 1, this nutrient combination induced proapoptotic CHOP, activated AMPK, and suppressed the reported therapeutic targets, such as ABCG2, GRP78, the side population, and YAP/TAZ. Compared to synthetic pharmacologic agents, combining cisplatin with edible dietary nutrients such as Se-Y and FO may offer therapeutic efficacy with minimal or no toxicity to the physiological system. The results suggest potential application of the Se-Y and FO combination to improve the outcomes of cisplatin-treated NSCLC with phenotypes such as A549 cells.

## Data Availability

Data are available on request.

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
