# Peer review of "Selenium Yeast and Fish Oil Combination Diminishes Cancer Stem Cell Traits and Reverses Cisplatin Resistance in A549 Sphere Cells"

_nutrients, 2022, doi:10.3390/nu14153232_

Round 1
Reviewer 1 Report
This is a nice article studying Cisplatin resistance in A549 cells/ spheroid. The main conclusion of this article is that Selenium Yeast and Fish Oil combination improves the Cisplatin activity on A549 spheroids, which shows more cancer stem cell properties compared to the A549 cells cultured as monolayers. The paper is good. However, I recommend a major revision.
1) The spheroid formation technique used in this article does not have any control over the spheroid size. There are already well-established methods of preparing uniform-sized spheroids using a microfluidic device or commercially available ULA plates. It would be much better if the authors could further verify the data with a size-controlled spheroid model.
2) The source and composition of the Selenium Yeast and Fish Oil should be described clearly (Line 134-138). Is the Fish oil prepared with a particular technique, or it is available commercially? Does the article describe the full form of DHA and EPA?
3) Figure 6 (B) describes the Se-Y 100 ng/mL+FO 100 uM in a combination of Cisplatin 20 ug/mL. Is it possible to check; what is the minimum concentration of Se-Y + FO that have a synergistic effect in combination with Cisplatin?
Author Response
This is a nice article studying Cisplatin resistance in A549 cells/ spheroid. The main conclusion of this article is that Selenium Yeast and Fish Oil combination improves the Cisplatin activity on A549 spheroids, which shows more cancer stem cell properties compared to the A549 cells cultured as monolayers. The paper is good. However, I recommend a major revision.
Response: Thanks a lot for reviewer’s helpful comments. We have tried our best to perform the additional experiments in the limited time frame and revised our manuscript as per your recommendations. Our replies are listed below and highlighted in blue.
1) The spheroid formation technique used in this article does not have any control over the spheroid size. There are already well-established methods of preparing uniform-sized spheroids using a microfluidic device or commercially available ULA plates. It would be much better if the authors could further verify the data with a size-controlled spheroid model.
Response: Thanks a lot for reviewer’s valuable instruction for preparing uniform-sized spheroids. However, it is unlikely to repeat all the experiments in the time frame of revision. For the uniform of the sphere cells used in this study, we have to perform additional requested experiments by sphere cells as described in current figures of this manuscript. The sphere cells used in this work reached a diameter of 100 μm (Figure 1A), the size which had been described in other studies of A549 (PLoS ONE 12(5): e0178286. 2017) and hepatoma (BMC Gastroenterology 2011, 11:71) sphere cells. In addition to the elevated CSC and EMT markers, we characterized our A549 sphere cells by analyzing their cisplatin resistance. Thanks again for the valuable instruction imparted by reviewer. In our next work, we will be glad to try the microfluidic device or commercially available ULA plates for preparing uniform-sized spheroids.
2) The source and composition of the Selenium Yeast and Fish Oil should be described clearly (Line 134-138). Is the Fish oil prepared with a particular technique, or it is available commercially? Does the article describe the full form of DHA and EPA?
Response: Thanks for the reviewer’s insightful comment. The selenium yeast used in this study is a mixture of small-molecule peptide-bound selenium extracted from the yeast grown in selenium-enriched medium. The fish oil used in this study is produced from anchovy by supercritical CO2 extraction. It is a mixture of natural triglyceride (TG) form of omega-3 fatty acids, which is much less artificial than that of ethyl ester (EE) or re-esterified triglyceride (rTG) form. We have revised the descriptions of Se-Y and FO in our revised version. The aforementioned selenium yeast and fish oil are commercially available from Do well Laboratories, Inc. and New Health Products Company. The full form of EPA (eicosapentaenoic acid), and DHA (docosahexaenoic acid) was described in Discussion of this article. In our revised version, the full form of EPA and DHA was added in the Materials and Methods.
3) Figure 6 (B) describes the Se-Y 100 ng/mL+FO 100 uM in a combination of Cisplatin 20 ug/mL. Is it possible to check; what is the minimum concentration of Se-Y + FO that have a synergistic effect in combination with Cisplatin?
Response: Thanks for reviewer’s helpful comment. We have performed additional experiments for this comment and inserted the results as the revised Figure 6C and Table 6. As shown in revised Table 6, the combination index (CI) values of combining Se-Y (25 ng/mL) and FO (25 µM) with cisplatin (10, 20, 40 µg/mL) were all below 1 (synergistic effect).

Reviewer 2 Report
Despite the good quality of scientific research, I cannot recommend this work. The main and fundamental disadvantage is the lack of in vivo studies. The main experiment in this work, in my opinion, should be the analysis of the effect of selenium yeast and fish oil on the growth and metastasis of subcutaneous xenografts in athymic nude mice. Also the gold standard in the CSC study is to determine the minimum dose of inoculation that is also missing. The next drawback is the use of only one cell line, which does not allow us to draw conclusions about the universality of the studied effect. If the authors do not have the opportunity to conduct in vivo studies, I recommend that they add at least one more lung cancer cell line to the studies.
Author Response
Despite the good quality of scientific research, I cannot recommend this work. The main and fundamental disadvantage is the lack of in vivo studies. The main experiment in this work, in my opinion, should be the analysis of the effect of selenium yeast and fish oil on the growth and metastasis of subcutaneous xenografts in athymic nude mice. Also the gold standard in the CSC study is to determine the minimum dose of inoculation that is also missing. The next drawback is the use of only one cell line, which does not allow us to draw conclusions about the universality of the studied effect. If the authors do not have the opportunity to conduct in vivo studies, I recommend that they add at least one more lung cancer cell line to the studies.
Response: Thanks a lot for reviewer’s insightful comments. Due to limited resources, we concede the disadvantages of lacking data from animal experiment and other lung cancer cell lines in this work. Therefore, we would not conclude the universality of the studied effect. Instead, we mentioned the name of cell line “A549” in our title and results and emphasized in the last sentence of our discussion that “combining Se-Y and FO may potentially improve the outcomes of cisplatin-treated NSCLC with phenotypes such as A549 cells”. To further highlight this issue and ignite additional needed research on this topic, we have revised the phrases in the end of discussion to “merit further tests in preclinical mouse models followed by clinical settings.”

Reviewer 3 Report
In the present study, the authors have investigated the effects of selenium yeast and fish oil combination on stemness and cisplatin sensitivity in A549 cell lines. This is an interesting and novel study, however, there are a few issues that need to be addressed by the authors:
1-The findings are limited to only 1 cell line. The authors need to expand their findings on two other lung cancer cell lines.
2-The applied concentration of FO is way too high, did the authors test its lower doses?
3-What is the synergism score in this combination therapy?
4-Could the authors repeat their apoptosis experiment with annexin V/PI staining?
5-Does the combination increase sensitivity to EGFR small molecule inhibitors such as erlotinib and gefitinib as well?
6-What is the biological replicate in this study? How many times each experiment was done?
7-In vivo data are required to validate these in vitro findings.
Author Response
Comments and Suggestions for Authors
In the present study, the authors have investigated the effects of selenium yeast and fish oil combination on stemness and cisplatin sensitivity in A549 cell lines. This is an interesting and novel study, however, there are a few issues that need to be addressed by the authors:
Response: Thanks a lot for reviewer’s insightful comments. Our replies are listed below and highlighted in blue.
1-The findings are limited to only 1 cell line. The authors need to expand their findings on two other lung cancer cell lines.
Response: Thanks a lot for reviewer’s insightful comments. Due to limited resources, we concede the disadvantages of lacking data from other lung cancer cell lines in this work. Therefore, we would not conclude the universality of the studied effect. Instead, we mentioned the name of cell line “A549” in our title and results and emphasized in the last sentence of our discussion that “combining Se-Y and FO may potentially improve the outcomes of cisplatin-treated NSCLC with phenotypes such as A549 cells”.
2-The applied concentration of FO is way too high, did the authors test its lower doses?
Response: Thanks for reviewer’s insightful comment. We have performed additional experiments for this comment and inserted the results as the revised Figure 6C and Table 6. The “Combination index (CI)” was used to evaluate the synergism between treatments. As shown in revised Table 6, the combination index (CI) values of combining Se-Y (25 ng/mL) and FO (25 μM) with cisplatin (10, 20, 40 μg/mL) were all below 1 (synergistic effect). As described in the Materials and Methods, each gram of FO contains 220 mg of DHA and 330 mg of EPA, and the concentrations of FO depicted in text and figures represent the content of omega-3 fatty acid (DHA and EPA). According to the plasma levels of DHA and EPA shown in the Table 2 of a paper by KURIKI ET AL. (J Nutr. 2003 Nov;133(11):3643-50.), the concentrations of fish oil used in our manuscript are clinically achievable.
3-What is the synergism score in this combination therapy?
Thanks for reviewer’s insightful comment. We have performed additional experiments for this comment and inserted the results as the revised Figure 3B, Figure 6C, Table 3 and Table 6. The synergism between treatments was evaluated by calculation of the “Combination index (CI)”. As shown in the revised Table 3 and Table 6, Most of the combination index (CI) values were all below 1 (synergistic effect), except a value of 1.078 in Table 3.
4-Could the authors repeat their apoptosis experiment with annexin V/PI staining?
Thanks for reviewer’s insightful comment. We have performed additional experiments and inserted the result as the revised Figure 2D.
5-Does the combination increase sensitivity to EGFR small molecule inhibitors such as erlotinib and gefitinib as well?
Thanks for reviewer’s insightful comment. In our previous report (Mar Drugs. 2020 Jul 29;18(8):399.), We have shown the effect of combining Selenium yeast and fish oil on reversing the acquired gefitinib resistance in HCC827 (EGFR E19 del) lung adenocarcinoma cells. However, we found these cisplatin-resistant A549 sphere cells (EGFR wild-type) are more sensitive to erlotinib and gefitinib than the parental A549 cells. We would like to further study this intriguing phenomenon and present it in our next report.
6-What is the biological replicate in this study? How many times each experiment was done?
Thanks for reviewer’s insightful comment. Two or three replicates were done and the representative one was shown in the figures.
7-In vivo data are required to validate these in vitro findings.
Response: Thanks a lot for reviewer’s insightful comments. Due to limited resources, we concede the disadvantages of lacking data from animal experiment in this work. We would like to present our interesting finding and ignite additional needed research on this topic. To highlight this issue, we have revised the phrases in the end of discussion to “merit further tests in preclinical mouse models followed by clinical settings.”.

Round 2
Reviewer 1 Report
All the questions from the round 1 review are addressed properly. Many thanks to the authors!
Reviewer 2 Report
To my regret, the authors could not make changes according to my comments, so I see no reason to change my verdict. However, if the journal editors consider the article level to be appropriate for the journal level, then the manuscript can be accepted as it stands.
Reviewer 3 Report
The authors have addressed my previous comments/questions and I have no further question to raise.